# New multimodal intervention to reduce irritable bowel syndrome (IBS) severity symptoms—Pilot study with a 12 month follow-up

**Maximilian Hinse**[1]*, **Anja Thronicke**[1,2], **Anne Berghöfer**[1], **Harald Matthes**[1,2,3]

**1** Institute of Social Medicine, Epidemiology and Health Economy, Charité–Universitätsmedizin Berlin, Corporate Member of Freie Universität Berlin and Humboldt-Universität zu Berlin, Berlin, Germany, **2** Research Institute Havelhöhe at the Hospital Gemeinschaftskrankenhaus Havelhöhe, Berlin, Germany, **3** Division of Gastroenterology, Infectiology and Rheumatology, Medical Department, Charité–Universitätsmedizin Berlin, Corporate Member of Freie Universität Berlin and Humboldt-Universität zu Berlin, Berlin, Germany

* maximilian.hinse@charite.de

**Data Availability Statement:** The anonymized data and all R-codes that support the findings of this study are openly available in the repository OSF (https://osf.io/fcbjh). DOI:10.17605/OSF.IO/FCBJH.

## Abstract

### Introduction

Irritable bowel syndrome (IBS) is characterized by patients' high level of suffering. Up to 60% of patients with IBS have symptoms of anxiety or depression and only little attention has been paid to their specific requirements. Anthroposophical multimodal therapy (AMT) has been shown to significantly improve health-related quality of life of patients with high symptomatic burden.

### Objective

The aim of this pilot study was to find out whether AMT meets the needs of IBS patients and the interactions of AMT with IBS, depression and anxiety

### Methods

Patients with diagnosed IBS were included in a feasibility study and received 12 sessions of AMT over 8 weeks (drks.de, DRKS00016890). The primary endpoint was the change of the IBS severity score (IBS-SSS) and changes were calculated by linear mixed effects analyses. The secondary endpoints were changes of self-reported PHQ-9 and GAD-7 for mental comorbidity as well as self-valued effectiveness and satisfaction of AMT.

### Results

Thirty-six patients, 89% female, were included in the study. AMT was successfully applied to IBS patients (-45 points in the IBS-SSS, p < .05). AMT had a large positive effect (-84 points in IBS-SSS, p < .003) on patients without anxiety or depression. Over time, patients with higher anxiety scores worsened with regard to their IBS compared to patients with depression and without mental comorbidity. The AMT effect was maintained at a 12 month follow up and both mentally affected and unaffected patients, had even lower IBS severity than shortly after AMT. AMT modules were rated by IBS patients as very effective.

**Funding:** The professorship of Harald Matthes (HM) and the research group Integrative and Anthroposophic Medicine (HM, MH and AT) at the Charité - Universitätsmedizin Berlin is funded by the Software AG foundation. The funders had no role in the study design, data collection and analysis, decision to publish, or preparation of the manuscript.

**Competing interests:** HM is a member of the board of directors of Weleda AG. The professorship of Harald Matthes (HM) and the research group Integrative and Anthroposophic Medicine (HM, MH and AT) at the Charité - Universitätsmedizin Berlin is funded by the Software AG foundation. The five young psychologists involved in this study were related to the Steinbeis University Berlin while writing their Master's theses. There are no other relationships/ conditions/circumstances that present a potential conflict of interest. The other authors have declared that no competing interests exist. For any other aspects of the submitted work no payment was received.

## Conclusion

Our findings suggest that an 8-week program of AMT improves the severity of IBS with an ongoing effect at a 12 month follow-up. Especially for patients without psychological comorbidities, AMT is very successful. Future IBS therapies should incorporate a modified multimodal concept with stronger psychological therapy modules in parallel for patients with depression and anxiety.

## Introduction

Irritable bowel syndrome (IBS) is one of the most frequently diagnosed gastrointestinal diseases [1] and, comprising around 50% of all visits to specialists, forms the largest diagnosis group in gastroenterology [2]. In Germany, approximately 10 million people suffer from IBS [3]. The prevalence in Germany is around 15–22% [4] and the pooled prevalence is 7% [5], with twice as many women being affected [5]. IBS patients are characterized by multiple, unexplained complaints that lead to a high level of suffering [4]. Between 20% and 49% of all IBS patients seek medical help, even several times, and cause both direct and especially indirect costs [4].

### Mental health and irritable bowel syndrome

IBS patients do not only suffer from somatic symptoms. Numerous studies point out that up to 60% of IBS patients suffer from serious psychological disorders, including in particular depression and anxiety disorders [6]. Compared to a healthy control group, IBS patients show significantly higher anxiety and depression scores [7]. Abdominal symptoms can influence anxiety and depression and at the same time psychological stressors influence pain perception [8, 9] and other physiological processes in the gut. This reciprocal connection between brain and gut is called the gut-brain axis [10–12] and, with the help of the biopsychosocial model of Tanaka et al. [13] can serve as an explanation for the complex interaction between biological, physiological and social factors [14, 15]. In the relationship between the intestine and the brain, signals from the intestine can be perceived by the enteric nervous system and transmitted to the brain through environmental influences such as nutrition, medication and infections, as well as being sent from the brain to the intestinal tract through the peripheral nervous system, e.g. stress, depression, anxiety. This mutual communication relationship is disturbed in IBS patients. In addition, the disturbed relationship between the intestine and the brain may increase anxiety, which in turn may increase patients' stress and keep them caught in a negative spiral of emotions [7, 16].

### Treatment of irritable bowel syndrome

In the German evidence and consensus based guideline [5] for the treatment of IBS, various and broad therapy measures are already recommended [5], including general information talks [2], gut hypnosis procedures [2, 17], dietary measures, psychotherapeutic procedures [2, 18–21], but also drug treatment such as probiotics [22, 23], dietary fibre [24] and antidepressants [2]. In the course of the multifaceted problems of the patients, the high level of suffering and the ambiguous origin or the undirected connection between the brain and the intestine, multimodal therapies are particularly suitable. In this way, the diverse possibilities of therapy can be combined to address both physical and psychological aspects at the same time. Both

sides of the gut-brain axis can be addressed by non-pharmacological therapies: on the physical level through exercise [25, 26], nutritional advice [22, 23, 27] and art therapy (painting and sculpting therapy) [25, 28, 29] and on the psychological level with the aid of psychoeducation, psychotherapy [18–21, 30], intestine hypnotherapy [2, 17, 31], cognitive restructuring [19] and also art therapies as nonverbal psychotherapy [25, 28, 29].

## Anthroposophic multimodal therapy

Anthroposophic medicine (AM) was founded at the beginning of the 20th century by Rudolf Steiner and Ita Wegman as an extension of the conventional medicine of the time. AM is a typical representative of integrative medicine, which integrates conventional and complementary therapeutic methods into an overall concept, taking into account the biological-living, psychosocial and spiritual dimensions of the human being in diagnosis and therapy [32]. The basis of AM is a positive concept of health [33] with the ability of the human being on the different levels to self-regulate (with hygio-, saluto- and autogenesis) [34]. According to the level of self-regulation, the various anthroposophical therapies are applied (hygiogenesis: phythotherapeutics, anthroposophics, manual therapies/rhythmic massages, baths, wraps, rubs, eurythmy etc.; salutogenesis: art therapies, psycho- and social therapies, etc.; autogenesis: biography work, meditation, etc.) and usually combined into so-called multimodal therapy concepts.

According to the anthroposophical disease concept, the functional intestinal diseases are seen as an imbalance of the human polarity of catabolic nerve-sense organization and anabolic metabolic activity in the digestive tract with dominance of increased consciousness processes in the metabolic area [25]. To restore the balance, therapeutic guidance is needed with inhibition of cognitive processes related to digestion or strengthening them against these cognitive processes. In conventional medicine, this connection between the nervous-sensory system and the digestive system is described by the construct of the so-called gut-brain axis [10–12]. This therapeutic support to balance the polar forces working in the human being is stimulated or directed with the help of special AM therapies (as listed above). The art therapies (speech, painting, music therapy and sculpting), for example, serve as a non-verbal dialogue within the patient himself; he is both the creator and the observer of his work and is thus stimulated to enter a self-reflective process. These therapies have been shown to be successful with anxiety disorders [29], depression and other chronic illnesses [28]. External applications such as rubs and wraps with oils are used to activate the vitality forces (hygiogenesis) and for relaxation and stimulation alike [32, 35, 36].

The therapy concept developed in this study combines therapies from AM (sculpting, painting, eurythmy and wraps) as well as already proven therapies for IBS, such as nutritional counselling, intestinal hypnosis and elements from behavioral therapy.

### Aim

The aim of the present study was to investigate a new anthroposophic multimodal short-term therapy (AMT) and its effect on IBS symptoms in a cohort of patients diagnosed with IBS and the interactions of AMT with IBS, depression and anxiety.

## Materials and methods

The study was conducted in cooperation with the Steinbeis University Berlin and the Max Lüscher foundation. Patients with diagnosed IBS were recruited by inviting them to an information event at the hospital Gemeinschaftskrankenhaus Havelhöhe (GKH) on 8th January 2018 via email distribution lists of the GKH, of medical care centers and of medical practices in the Berlin-Brandenburg region. Patients of both gender between 18 and 90 years old were

included. The diagnosed IBS had to be confirmed by Rome IV [37] criteria. Exclusion criteria were lack of diagnosis, suicidal tendencies, psychotic experiences and participation in other studies. Drug therapy for IBS had to be stable or without effect for 6 weeks.

## Design

The study was conducted as a feasibility study with a repeated measures pretest-posttest design. Originally, a waiting group design was planned. It became apparent during recruitment that the enrolment of this patient group was not so easy for this setting of therapy. The groups in the evening, for example, were less well received than those in the afternoon. Here it was not possible to offer more or larger groups earlier in the day due to the ongoing hospital operations and the people involved working in the medical institutions. The primary outcome IBS-SSS and the secondary outcomes PHQ-9 and GAD-7 were measured at the first day of intervention (baseline), post AMT at the last day of intervention and after a 12 month follow-up. The AMT PREM questionnaire was collected only at the last day of AMT with a small follow-up specific questionnaire after 12 months. All included participants were contacted 12 months after posttest for their follow up participation in April 2019.

## Anthroposophic multimodal treatment (AMT)

All participants received an 8-week lasting AMT with 12 AMT sessions between 16th January 2018 and 4th April 2018. In the first four weeks, there were two AMT appointments each week. In the following four weeks, there was only one appointment per week. Participants could choose a group with appointments in the afternoon or evening. A total of four groups were offered. Two groups started immediately on 16th January and two groups only after a four-week waiting period. The AMT appointments lasted two hours each and were divided into four 30-minute sessions. The AMT consisted of the following modules that took place in the rooms of the hospital GKH:

- The aim of *Psychoeducation* was to promote patients' understanding of their illness, to promote self-responsibility in dealing with the disease, to support coping with the disease and to reduce fears and feelings of guilt. Psychoeducation was conducted by two psychologists in six sessions.

- Art therapies: The aim of *painting therapy* and *sculpting* was to promote and support patients' introspective self-regulation processes and attention redirection. Both were conducted by two experienced art therapists of the GKH in four sessions each.

- Movement therapy (*eurythmy therapy*) was used to stimulate intentionality and the flow of movement as well as to achieve improved body awareness and body control to experience both tension and relaxation in equal measure. Eurythmy was conducted by an experienced therapist in six sessions.

- The aim of *External application* was to teach self-help strategies with knowledge of different oils for external application as a wrap for home use to support self-healing processes and was conducted by two psychologists in four sessions.

- *Healing imagination*, intestinal hypnotherapy was conducted by two psychologists in eight sessions to provide patients peace and relaxation.

- The aim of *Nutritional counselling* was to give patients education about food intolerances and different food groups and their impact on their microbiota and IBS. The experienced

nutritionist did not give advice on specific diets but gave education and answered individual questions in four sessions.

- The group for *cognitive training* with the aim of achieving improved reflection on the body states and changes in feelings that have been altered in the therapies was conducted by psychologists at the end of all 12 sessions.

The AMT modules were conducted either by young psychologists under supervision or by experienced therapists from the hospital GKH. The cooperation of the young psychologists took place during their studies at Steinbeis University as part of their Master's thesis and their practice was supervised by experienced senior professionals.

## Questionnaires

The general medical history questionnaire was used to obtain demographic data (age, gender, family, occupation) as well as other questions about IBS symptoms and how they were managed and what kind of medical help had been used to date. The primary endpoint of the study, the IBS severity score (IBS-SSS) was measured with the validated Irritable Bowel Syndrome— Severity Scoring System (IBS-SSS) by Francis, Morrel and Whorwell [38]. Five different items were answered on a visual analogue scale and result in the irritable bowel syndrome severity score (from min. 0 to max. 500 points). According to the validation study by Francis, Morrel and Whorwell [38] the results can be interpreted as follows: Scores below 75 mean "in remission", between 75 and 174 were considered "mild disease", 175–299 as "moderate" and 300 or higher, as "severe" disease. Minimum Clinically Important Difference (MCID) is reported at a change of 50 points or more and is considered a treatment responder. In this study, the German version of the IBS-SSS by Betz et al. [39] was used. The patients' comorbidities were recorded as secondary endpoints. The validated "Patient Health Questionnaire" (PHQ-9) by Kroenke and Spitzer [40] was used to diagnose depressive disorders. The PHQ-9 measures the severity of a depressive disorder with nine questions resulting in a sum score between 0 and 27 points. A score of 5 or more indicates mild depression, from 10 moderate, from 15 moderate-severe and above 20 severe depression. The "General Anxiety Disorder questionnaire" (GAD-7) by Spitzer et al. [41, 42] is used to diagnose people with generalised anxiety disorder. The seven questions of the GAD-7 result in a sum score that can reach values between 0 and 21 points. The result of the total score can be divided into three degrees of severity: from 5 points corresponds to a mild anxiety disorder, from 10 points to a moderate anxiety disorder and from 15 points or more to a severe anxiety disorder [43]. A severity level above 10 points in the PHQ-9 and above 10 points in the GAD-7 represents moderate depression or moderate anxiety symptoms. This cut-off was used to divide the patients of the study into two groups, one group with anxiety or depression symptoms (mental comorbidity) and one group without comorbidity (no mental comorbidity).

A newly developed questionnaire on self-reported effectiveness and satisfaction with anthroposophic complex treatment was used as a further secondary endpoint. This questionnaire served as a first test for a later version to be developed into a patient-reported experience questionnaire (AMT PREM) for the evaluation of anthroposophic complex treatments from the patient's perspective. The questionnaire contains a general and a specific part for each of the eight therapy modalities.

## Statistical analysis

A sample size calculation was not performed due to the pilot character of the study. All results were considered exploratory. In addition to descriptive analyses of the study data, student's t-

test was used for the pure mean comparisons of the baseline and post AMT results. Welch's two sample test was used for comparisons between mental comorbidity groups. Results were presented descriptively in Table 2 and between group comparisons for IBS and IBS with mental comorbidity groups. All outcomes were reported with pre-, post- and follow-up results with false discovery rate correction for multiple testing (q value). Furthermore, two linear mixed effect model analyses were performed for the main analysis with results presented in Table 3. We fitted a linear mixed model to predict the primary outcome IBS-SSS with AMT, PHQ-9 and GAD-7 scores as fixed effect (all three at baseline, post-treatment and at 12 month follow-up). The model included participant's ID as a random effect [44–46]. Two models were performed to better differentiate the effect of interaction (model 2) from the main effect (model 1). In order to investigate the influence of different covariates, such as depression or anxiety disorder on the success of the AMT, these were included as covariates in the multivariate mixed effect analyses. Fit measures were also reported in Table 3 (AIC, BIC, RMSE, Sigma and ICC). All common statistical assumptions were carefully addressed for our outcomes, e.g. linearity, homogeneity of variances, analysis of normal distribution of residuals and analysis of outliers; using analysis plots and analysing results [44–47]. In our main analysis, a linear mixed model was calculated in which the assumptions of regressions also apply and were therefore taken into account. Other assumptions, such as the assumption of the independence of the errors or the homogeneity of the regression slopes, were balanced by multilevel models [38].

Analyses were conducted using the R Statistical language (version 4.1.0; R Core Team, 2021 [48]) with RStudio Version 1.4.1717 on macOS 12.0.1, using the most recent versions of R packages: tidyverse [49], lme4 [50], ggstatsplot [51], ggeffects [52], sjPlot [53], psych [54], gtsummary [55]. All data and R code is available at the OSF repository [56]. A 5% significance level was set for the statistical analyses.

The qualitative open questions in the questionnaires were interpreted by a quantitative count of the most frequent answers and additionally analysed using a latent dirichlet allocation analysis (LDA) described by Blei, Ng and Jordan [57] with the R packages tidytext [58] and topicmodels [59] in the most recent versions.

## Ethics approval, trial registration and consent to participate

The study was granted ethics approval by the ethics Committee of the Berlin Medical Association (Eth-35/17). All study participants gave written informed consent to participate in the study. The study took place in compliance with professional regulations, the Declaration of Helsinki and the recommendations of the ICH Guideline for Good Clinical Practice. The study is retrospectively registered in the German trial registry DRKS (drks.de, DRKS00016890). The study was initiated and conducted as a pilot study with the help of four young psychologists who completed their master's theses. Due to limited resources, the study was only registered retrospectively. The authors confirm that all ongoing and related trials for this intervention are registered.

## Results

### Sample description

A group of 51 patients with confirmed IBS diagnosis were included in the study. Forty-four of them appeared at the first visit, 36 participated in the AMT and 21 patients returned the follow-up questionnaire (Fig 1). One patient did not send back the post questionnaire and 14 patients were lost to follow-up, see Fig 1.

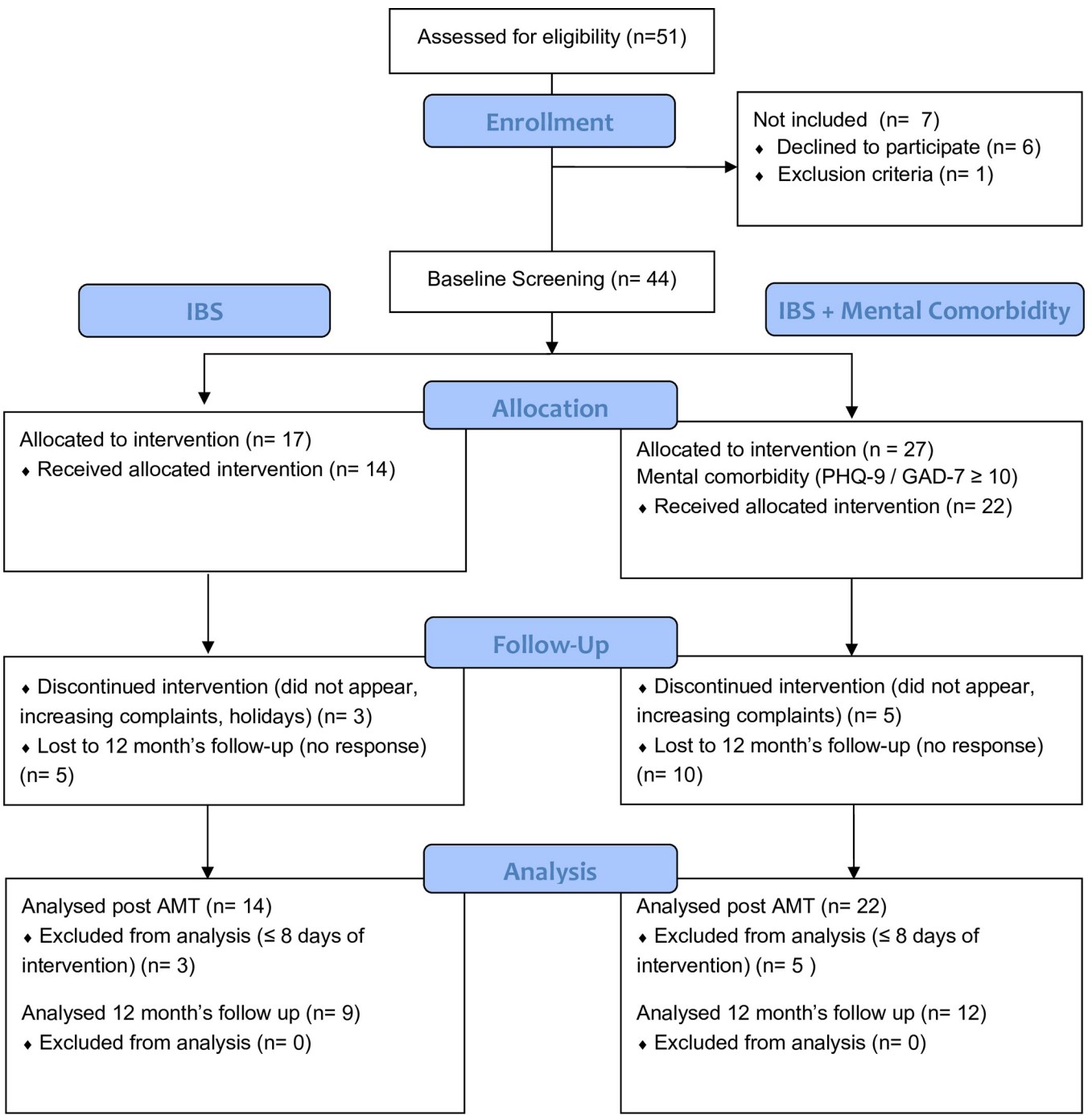

**Fig 1. Flow chart of the study.**

The mean age of the study cohort was 53.2±14.6 with 89% of female gender. Forty-four percent of the participants had a higher educational degree with 19% of them having an academic degree, see Table 1. One half of the participants (53.7%) were employed and the other half consisted mainly of pensioners, job seekers and people undertaking education, see Table 1. 78% of the study group had other medical, 33% psychological, 28% physiotherapy and 14% homeopathic treatment. 14% had no other treatment. More than half of the participants had children

**Table 1. Sample description of participants.**

| Characteristics | Overall, N = 36[1] | % |
|---|---|---|
| | n | |
| Sex | | |
| female | 32 | 89 |
| male | 4 | 11 |
| Age, mean (min-max), years | 53 | 21, 88 |
| Education | | |
| university degree | 7 | 19 |
| high school graduation | 14 | 39 |
| secondary school certificate | 9 | 25 |
| skilled occupation | 6 | 17 |
| Profession | | |
| employed | 12 | 34 |
| teacher | 5 | 14 |
| independent | 2 | 6 |
| pensioner | 9 | 26 |
| student | 2 | 6 |
| unemployed | 5 | 14 |
| unknown | 1 | 3 |
| Other treatment | | |
| medical treatment | 28 | 78 |
| psychological treatment | 12 | 33 |
| physiotherapy treatment | 10 | 28 |
| homeopathic treatment | 5 | 14 |
| no other treatment | 5 | 14 |
| Children | | |
| 0 | 17 | 47 |
| 1 | 9 | 25 |
| 2 | 8 | 22 |
| 3 | 2 | 6 |

(53%). A total of 28 persons stated that they were currently undergoing further medical treatment. The reasons given for the treatment were, for example: irritable bowel syndrome (10), stomach (5) or intestinal (5) complaints, Hashimoto's disease (3), or depression (3).

## Effect of AMT on IBS severity and mental comorbidity

The group of patients in this study had an overall moderate IBS severity score at baseline (mean: 287 ± 65) and 8 weeks post-AMT (mean: 241 ±104), as illustrated in Table 2 and the mean difference suggests a statistically significant medium positive effect (mean difference = 45.74, 95% CI [16.55, 74.93], t(34) = 3.18, p = 0.003, Cohen's d = 0.54). The difference of 45.74 was below the clinical relevant difference of 50 points [38, 39]. At the 12 month follow-up, IBS severity had reduced even more, by 38 points from post AMT to a total of 203 (±107), but again not clinically relevant.

Patients of the total study cohort had an overall moderate depressive symptomatology at baseline as measured by the PHQ-9 [42] below the cut-off for moderate-severe depressive symptomatology (mean PHQ_pre: 9.9 ±4.0). At the end of the AMT, patients continued to have depressive symptomatology at a moderate level (mean PHQ_post: 8.1±4.1). The

**Table 2. Severity, depression and anxiety scores among IBS patients, pre, post AMT and at 12 month follow-up.**

| Characteristics | N | Overall | IBS | IBS + Mental comorbidity[*] | Diff.[1] | 95% CI[2] | p-value[1] | q-value[3] |
|---|---|---|---|---|---|---|---|---|
| | | | N (%) | N (%) | | | | |
| **Sex** | 36 | | 14 (39) | 22 (61) | | | | |
| female | 32 | 32 (89%) | 13 (93) | 19 (86) | | | | |
| male | 4 | 4 (11%) | 1 (7.1) | 3 (14) | | | | |
| | | | Mean (SD) | Mean (SD) | | | | |
| **Age** | 36 | 53 (15) | 54 (19) | 53 (11) | 0.92 | -11, 13 | 0.9 | >0.9 |
| **IBS-SSS pre** | **36** | **287 (65)** | **267 (70)** | **299 (59)** | **-32** | **-78, 15** | **0.2** | **0.3** |
| IBS-SSS 1b: severity of abdominal pain | 36 | 40 (24) | 34 (28) | 44 (20) | -9.7 | -28, 8.4 | 0.3 | 0.3 |
| IBS-SSS 1c: frequency of abdominal pain | 36 | 51 (38) | 44 (44) | 56 (34) | -13 | -41, 16 | 0.4 | 0.4 |
| IBS-SSS 2b: severity of abdominal distension | 36 | 53 (24) | 55 (21) | 52 (26) | 2.5 | -13, 18 | 0.7 | 0.8 |
| IBS-SSS 3: satisfaction with bowel habit | 36 | 67 (18) | 67 (14) | 66 (21) | 0.55 | -11, 12 | >0.9 | >0.9 |
| IBS-SSS 4: impairment in life | 36 | 76 (14) | 68 (16) | 81 (11) | -12 | -23, -2.0 | 0.021 | 0.047 |
| **IBS-SSS post** | **35** | **241 (104)** | **183 (111)** | **280 (81)** | **-97** | **-169, -25** | **0.010** | **0.029** |
| IBS-SSS 1b post: severity of abdominal pain | 35 | 35 (25) | 20 (28) | 44 (19) | -24 | -42, -6.8 | 0.009 | 0.029 |
| IBS-SSS 1c post: frequency of abdominal pain | 35 | 44 (34) | 27 (36) | 56 (28) | -29 | -52, -5.4 | 0.018 | 0.045 |
| IBS-SSS 2b post: severity of abdominal distension | 35 | 47 (24) | 38 (28) | 53 (20) | -15 | -33, 3.4 | 0.10 | 0.2 |
| IBS-SSS 3 post: satisfaction with bowel habit | 35 | 52 (22) | 47 (23) | 56 (22) | -8.9 | -25, 7.0 | 0.3 | 0.3 |
| IBS-SSS 4 post: impairment in life | 35 | 63 (22) | 51 (22) | 71 (18) | -20 | -34, -5.5 | 0.009 | 0.029 |
| **IBS-SSS 12 Month Follow-Up** | **21** | **203 (107)** | **159 (128)** | **235 (79)** | **-77** | **-182, 29** | **0.14** | **0.2** |
| **PHQ-9 pre** | 36 | 9.9 (4.0) | 5.9 (2) | 12.4 (2) | -6.6 | -8.2, -4.9 | <0.001 | <0.001 |
| **PHQ-9 post** | 35 | 8.1 (4.1) | 4.2 (2) | 10.6 (3) | -6.4 | -8.1, -4.7 | <0.001 | <0.001 |
| **PHQ-9 Follow-Up** | 20 | 9.2 (5.5) | 6.4 (5) | 11.4 (5) | -4.9 | -9.8, -0.07 | 0.047 | 0.085 |
| **GAD-7 pre** | 35 | 8.2 (4.9) | 4.9 (2) | 10.4 (5) | -5.5 | -8.0, -2.9 | <0.001 | <0.001 |
| **GAD-7 post** | 36 | 7.4 (4.5) | 4.9 (3) | 9.1 (5) | -4.2 | -6.8, -1.7 | 0.002 | 0.010 |
| **GAD-7 Follow-Up** | 21 | 9.4 (6.0) | 6.2 (5) | 11.8 (6) | -5.6 | -11, -0.63 | 0.029 | 0.059 |

[1]Welch'stwo sample t-test

[2]CI = confidence interval

[3]false discovery rate correction for multiple testing

[*]Cut-off criteria for 'Mental Comorbidity' is defined as $\geq$ 10 points in either PHQ-9 (depression) or GAD-7 (anxiety) at baseline

difference was statistically but not clinically relevant (mean difference = 1.77, 95% CI [0.93, 2.62], t(34) = 4.26, p < .001, Cohen's d = 0.72, see Table 2).

A similar picture emerges for anxiety symptoms, measured with the GAD-7. Patients had mild anxiety symptoms both at the beginning of the study (mean GAD_pre = 8.2±4.9 and at the end of the AMT (mean GAD_post = 7.4±4.5, see Table 2). At the 12-month follow-up, patients' mean PHQ-9 and GAD-7 scores remained at similar levels to baseline, at post AMT and 12 month follow-up (Mean GAD_fu = 9.2±5.5; Mean PHQ_fu = 9.4 ±6.0, see Table 2).

## Effect of mental comorbidity (depression / anxiety) on IBS severity

A total of 22 persons (61%) had a mental comorbidity and 14 persons had none. Of these, all the 22 comorbidity participants had depressive symptoms and 11 of these had additional anxiety symptoms, see Table 2. By analyzing the results of the IBS severity score separately for each patient group, a large IBS-SSS difference between both groups for post-AMT was found, see Fig 2 (mean in IBS group = 183.14, mean in the IBS + comorbidity (anxiety or depression) group = 280.14). The results suggest a statistically significant and large negative effect (difference = -97.00, 95% CI [-168.54, -25.46], t(22.01) = -2.81, p = 0.010, Cohen's d = -1.20), see

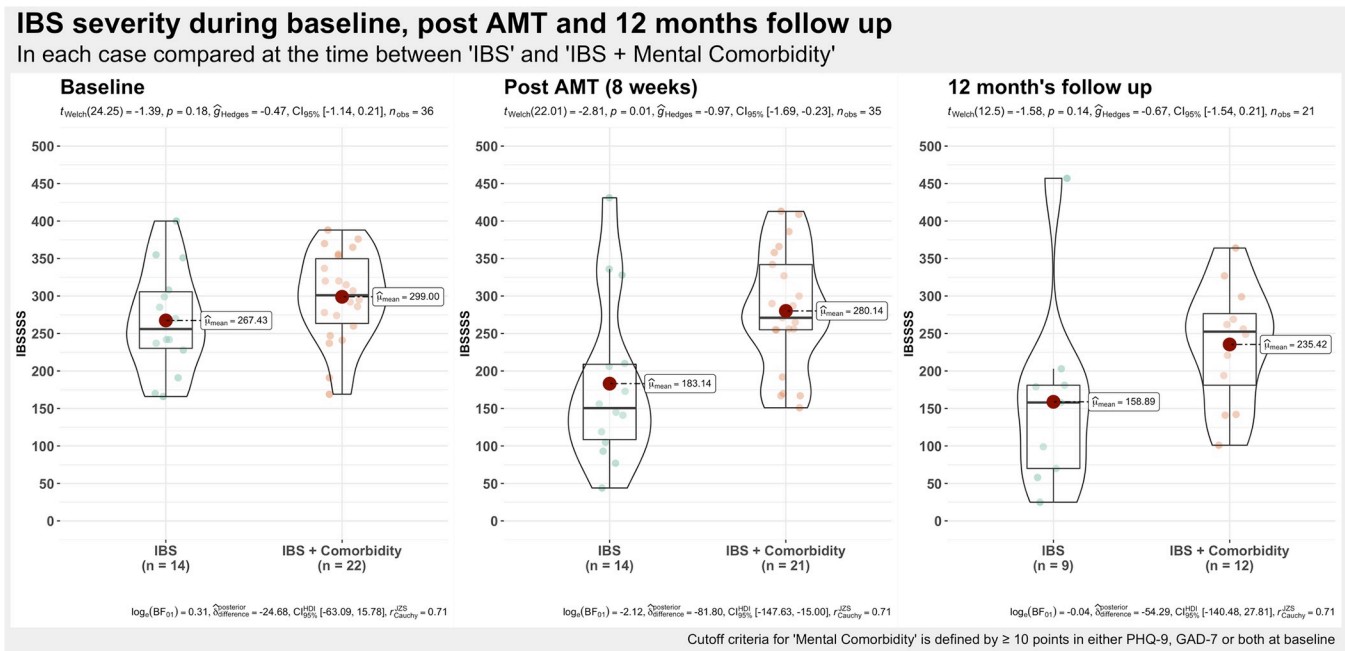

**Fig 2. IBS severity syndrome at baseline, post-AMT and follow-up for both groups: No mental comorbidity and with mental comorbidity (depression and anxiety).**

Table 2. At the 12 month follow-up, both groups improved their severity scores but also converged at the same time (mean in the no mental comorbidity group = 158.89, mean in the mental comorbidity group = 235.42). At follow-up, both groups had no statistically significant difference anymore (difference = -77, 95% CI [−182, 29], t(12.50) = -1.58, p = 0.139), see Fig 2. One patient has been identified as an outlier at follow-up in the IBS group without mental comorbidity. Having a closer look at this patient's data, the scores changed from baseline = 308 to post AMT = 328 and at follow up to 457. This patient is the only one who is identified as an outlier at post AMT and with an even greater percentage at follow-up. When this patient's scores are excluded from the analysis, mean scores of IBS-SSS in the IBS-only group without depression or anxiety symptomatology change dramatically from 183.14 at post AMT to 172.0 and with the largest effect at follow up from 158.89 to 121.62. The difference between the IBS-only group and IBS with mental comorbidity at follow-up is statistically significant with 113.8 points difference in IBS-SSS score. The difference is more than two times the clinically relevant amount of >50 points in IBS-SSS. Nevertheless, this patient's data is still included in the following statistical analysis in order not to influence the results of this small sample too drastically.

Of the 22 patients with mental comorbidity, 10 also received psychotherapeutic treatment (see Table 1). Comparing the post-AMT difference in IBS-SSS between patients with mental comorbidity (depression or anxiety), patients without psychotherapeutic treatment (mean = 267.75) and patients receiving psychotherapy (mean = 296.67) suggests a small negative effect (difference = -28.92, 95% CI [-103.59, 45.76], t(18.01) = -0.81, p = 0.427, Cohen's d = -0.38).

To analyse the effectiveness of the AMT, we fitted two linear mixed models by predicting IBS severity by AMT, depression and anxiety. The first model revealed a fixed effect of post-AMT as statistically significant and negative (marginal $R^2$ = .20 and conditional $R^2$ = .66, see Table 3), predicting a reduced IBS-SSS (beta = -41.12, p = 0.006, see Table 3). Furthermore, it

**Table 3. Comparing AMT treatment effect on IBS severity score post-AMT and at 12 month follow-up with interaction effects of depression and anxiety with treatment on IBS severity scores in linear mixed effects analysis.**

| Parameter—fixed | Model 1 | | | | | Model 2 | | | | |
| | without interaction | | | | | with interaction effects | | | | |
| | Coefficient | 95% CI | p | Std. Coef. | Std. Coef. 95% CI | Coefficient | 95% CI | p | Std. Coef. | Std. Coef. 95% CI |
|---|---|---|---|---|---|---|---|---|---|---|
| (Intercept) | 225.43 | (173.51, 277.35) | <**0.001** | 0.35 | (0.05, 0.65) | 231.90 | (165.23, 298.57) | <**0.001** | 0.30 | (0.003, 0.59) |
| Treatment [post] | -41.12 | (-70.01, -12.22) | **0.006** | -0.42 | (-0.72, -0.13) | -93.01 | (-157.95, -28.06) | **0.006** | -0.31 | (-0.60, -0.03) |
| Treatment [follow-up] | -78.82 | (-113.11, -44.53) | <**0.001** | -0.81 | (-1.16, -0.46) | -85.44 | (-158.71, -12.18) | **0.023** | -0.84 | (-1.18, -0.50) |
| PHQ-9 | 6.37 | (0.28, 12.45) | **0.041** | 0.29 | (0.01, 0.57) | 11.72 | (3.12, 20.32) | **0.008** | 0.53 | (0.14, 0.92) |
| GAD-7 | 0.25 | (-5.21, 5.71) | 0.928 | 0.01 | (-0.27, 0.30) | -6.94 | (-13.84, -0.03) | **0.049** | -0.36 | (-0.72, -0.001) |
| Treatment [post] *PHQ-9 | | | | | | -2.92 | (-12.80, 6.96) | 0.558 | -0.13 | (-0.58, 0.32) |
| Treatment [follow-up] * PHQ-9 | | | | | | -13.48 | (-25.81, -1.15) | **0.033** | -0.61 | (-1.17, -0.05) |
| Treatment [post] * GAD-7 | | | | | | 10.76 | (2.11, 19.41) | **0.015** | 0.56 | (0.11, 1.01) |
| Treatment [follow-up] *GAD-7 | | | | | | 15.03 | (4.60, 25.46) | **0.005** | 0.78 | (0.24, 1.33) |
| **Random Effects** | | | | | | | | | | |
| | Model 1 | | | | | Model 2 | | | | |
| ID–random | 66.03 | | | | | 64.26 | | | | |
| Residual–random | 56.73 | | | | | 52.98 | | | | |
| ICC | 0.58 | | | | | 0.60 | | | | |
| N | 36 ID | | | | | 36 ID | | | | |
| RMSE | 45.85 | | | | | 41.21 | | | | |
| Sigma | 56.73 | | | | | 52.98 | | | | |
| AIC | 1003.89 | | | | | 980.50 | | | | |
| BIC | 1021.31 | | | | | 1007.88 | | | | |
| Observations | 89 | | | | | 89 | | | | |
| Marginal $R^2$ / Conditional $R^2$ | 0.203 / 0.661 | | | | | 0.288 / 0.712 | | | | |

revealed a statistically significant negative IBS-SSS follow-up effect, predicting an even more reduced severity (beta = -78.82, p < .001,). The PHQ-9 follow-up effect was statistically significant and positive (beta = 6.37, p = 0.041). The follow-up GAD-7 effect was statistically non-significant and positive (beta = 0.25, p = 0.928). While both depression and anxiety scores were decreasing between baseline and post- AMT (see Table 2), both scores increased during the 12 month follow-up, but were not clinically or statistically relevant in any group.

To investigate the interaction effects between PHQ-9, GAD-7 and AMT as well, both interactions were included within a second mixed effects analysis (model 2), indicated by the right three columns of Table 3. The second model's total explanatory power is substantial (conditional $R^2$ = .71) and the part related to the fixed effects alone (marginal $R^2$) was .29 (see Table 3). The model's intercept was 30.45 (95% CI [1.84, 59.05]). This second analysis revealed a statistically significant and negative fixed effect of post-AMT (beta = -31.22, p = 0.028). The effect of follow-up AMT was statistically significant and negative (beta = -83.88, p < .001). The effect of PHQ-9 was statistically significant and positive (beta = 11.72, p = 0.008) and the effect of GAD-7 in this model was also statistically significant and negative (beta = -6.94, p = 0.049).

The second model shown in Table 3 and Fig 3 additionally included interaction effects. While the interaction effect of PHQ-9 (depression score) with treatment [post] was statistically

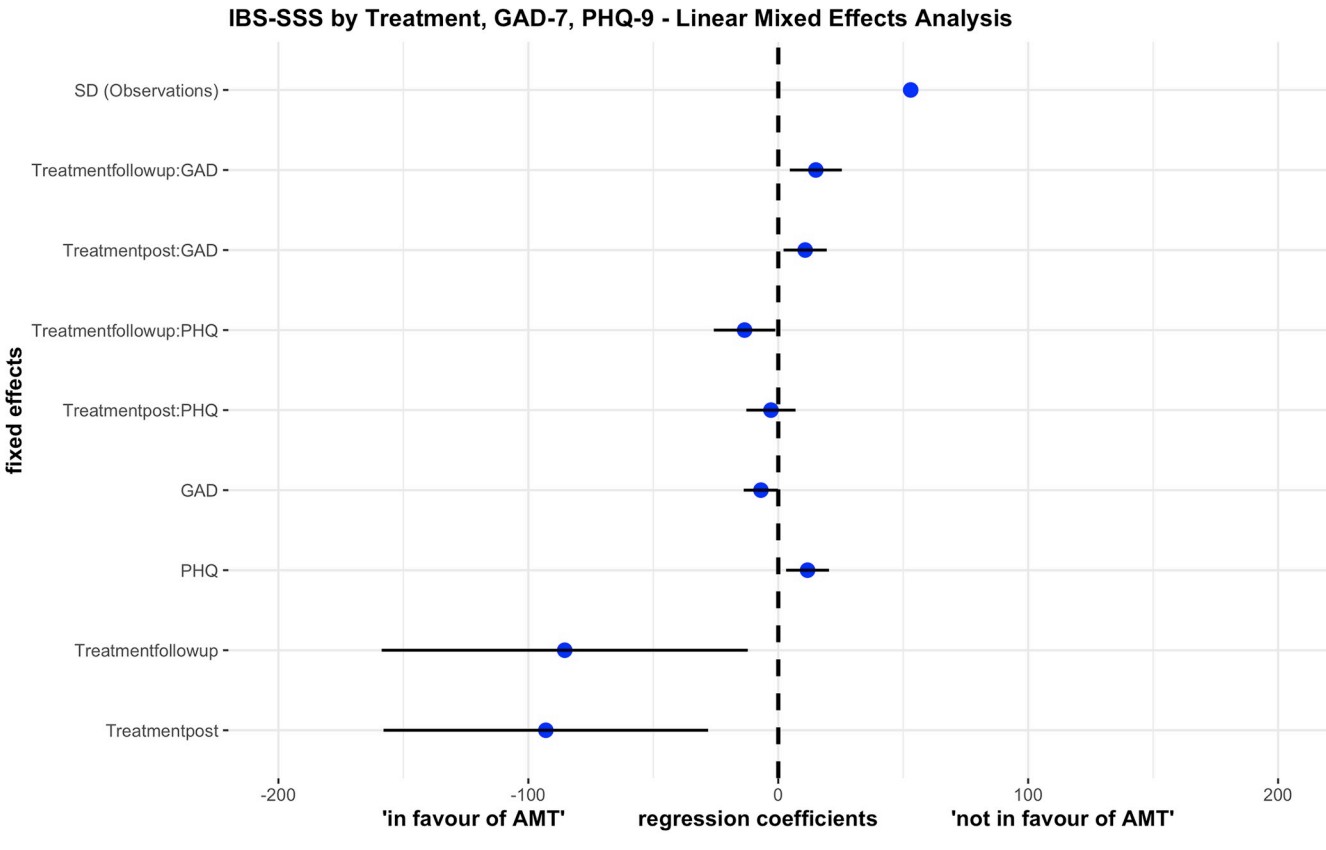

**IBS-SSS by Treatment, GAD-7, PHQ-9 - Linear Mixed Effects Analysis**

AIC = 981, BIC = 1008

**Fig 3. Forest plot showing the linear mixed effect analysis, predicting IBS severity by treatment, depressive and anxiety symptomatology and interaction effects between AMT at post and follow-up and mental comorbidity at both post and follow-up.** Negative regression coefficients predict reduced IBS and positive coefficients predict increased IBS severity.

non-significant and negative (beta = -2.92, p = 0.558), the interaction effect of PHQ with treatment [follow-up] was statistically significant and negative (beta = -13.48, p = 0.033). Furthermore, the interaction effect of GAD-7 (anxiety score) with treatment [post] and [follow-up] was statistically significant and positive, respectively (beta = 10.76, p = 0.015; beta = 15.03, p = 0.005). While the effect of PHQ-9 is negative and thus predicting a reduced IBS severity score at post treatment, this interaction has a significant effect only at the 12 month follow-up and PHQ-9 scores had no effect at post-treatment. Patients with depressive symptomatology had a decreased IBS severity only at the 12 month follow-up. The interaction effects of GAD-7 scores for post-treatment and for the 12 month follow-up were both positive and significant, meaning that with increasing GAD-7 scores the IBS severity scores were as well increasing at post-AMT and at the 12 month follow-up.

## Perceived satisfaction and effectiveness with AMT

The study participants were asked about their subjective satisfaction and their impression of effectiveness with AMT [56]. Twenty-three out of 36 patients (64%) rated the intevention modules and were very satisfied with the AMT procedure with up to 96% satisfaction. Eurythmic therapy and cognitive trainingmodules were rated slightly less positively with 65% and 70% satifcation, respectively. Overall, the individual AMT modules were perceived as very

effective (agreement between 78% and 92%). Only the eurythmy (56%) and cognitive training (60%) modules were rated slightly more indifferently. All other modules were rated to be even more effective (between 78% and 96%). Comparing the AMT modules of painting, imagination and sculpting, the agreement values of comorbid persons with GAD-7 or PHQ-9 revealed a slightly higher satisfaction than in the group without mental comorbidity [56].

## Discussion

The results of the present study reveal that an 8-week AMT effectively reduced IBS symptom severity. While the 8-week AMT only moderately reduced IBS severity in the total cohort, it reduced IBS severity for patients without mental comorbidity on a clinically relevant level. The improvement was clinically highly relevant and statistically significant compared to patients with mental comorbidity (depression and anxiety). This effect was maintained and further progressed during the 12 month follow-up for patients without mental comorbidity. These patients had an additional moderate level IBS severity reduction in the 12 months after AMT. If the one patient identified as an outlier was excluded from the calculations, a reduction of 113.8 points in severity would also occur at the 12 month follow-up. The difference is more than two times the clinically relevant amount of >50 points in IBS-SSS. In addition, we could show that the AMT modules implemented in this study had a delayed sufficient effect even for IBS patients with mental comorbidity of depression but no direct nor delayed effect for patients with symptoms of anxiety. The mixed effects analysis even showed a statistically significant effect for anxiety symptomatology to worsen the outcome of IBS severity over time, directly after AMT and over the 12 month follow-up period.

Several studies have revealed that cognitive behavioral therapy (CBT) is effective for patients with IBS but did not differentiate between IBS patients without mental comorbidity and patients with depression or anxiety symptomatology. The AMT did not include psychotherapy and it seems that the psychotherapeutic elements of the AMT concept (art therapy and cognitive training group) are either not sufficient or the duration of an 8-week AMT for IBS patients with mental comorbidity symptoms is too short.

Recent studies show that there are different ways to effectively address different aspects of severity in IBS patients. Gut hypnotherapy, nutritional interventions, or CBT implemented as a single session, group therapy, by telephone or video calls can, for example, effectively help to reduce IBS severity [2, 18, 19, 24, 30, 60]. The concept of combining successful therapeutic modules into a multimodal AMT therapy concept, each addressing different aspects of IBS severity, proves to be successful in the present pilot study.

The results of the present study reveal that the frequency of mental comorbidity in the IBS cohort was around 61%, reflecting the prevalence indicated by other studies [7, 16, 61]. Due to the high prevalence of psychological comorbidities in IBS patients, it is obvious that many therapeutic approaches in other studies focus on both medication-based therapy with antidepressants and non-medication-based therapy with different variants of behavioural therapy concepts [18, 19, 30, 60, 62]. In these studies, many different concepts were implemented and were more or less helpful in reducing the severity of irritable bowel syndrome. Antidepressants, CBT, dietary programs and other therapies are comparable to each other in their effectiveness, whereas hypnotherapeutic programs appear to be more successful in the treatment of IBS patients [19]. To the best of our knowledge, however, the group of IBS patients without psychological comorbidity has been neglected in previous studies and not been addressed by further differentiated treatment concepts. Due to the high prevalence of psychological comorbid patients, these studies focused primarily on the additional comorbidity while neglecting patients without existing mental comorbidity. A meta-analysis showed that integrative therapy

and cognitive behavioral therapy showed superior efficacy in reducing IBS severity when defining efficacy as >50% reduction of GI symptoms compared to standard drugs [63]. Integrative therapies such as multimodal anthroposophical therapies and especially art therapies (painting and sculpting) are used as a kind of non-verbal psychotherapy and have an anxiety alleviating effect, among other effects [28, 29].

Anxiety and depression were shown in the present study to be associated with higher IBS severity. These findings correlate with earlier studies indicating that the prevalence of these comorbidities in IBS patients is twice as high compared to the general population [7, 16, 61]. In our study, we observed that the 8-week AMT had no significant effect on IBS severity in patients with depressive or anxious symptomatology. However, during the 12 month follow-up, IBS severity significantly improved in patients with depressive symptomatology but not in patients with additional anxiety symptomatology where IBS severity even worsened after 8 weeks of AMT and at the 12 month follow-up. This delayed effect in depressive IBS patients could be explained by the fact that they need more time to adapt to new changes (e.g. AMT treatment) compared to IBS patients without mental comorbidity. Additional psychotherapeutic treatment after AMT must be considered for IBS with depressive comorbidity for the long-term effect. An explanation of the long-term effect as regression to the mean is very unlikely, since on the one hand the IBS symptoms existed for a long time before inclusion in the study and on the other hand a regression to the mean should already have been recognizable after 6 months.

In IBS patients with anxiety symptomatology, the observed deterioration of IBS severity during AMT and follow-up may also be triggered by worsened IBS or even anxiety symptomatology leading to increased IBS severity [7, 16]. Considering anxiety separately as shown by the linear mixed effect regression, it was a hindrance to a positive IBS progression at post-AMT and in particular at the 12 month follow-up. The interaction effect in the linear mixed effect analysis was very impressive. At the 12 month follow-up, patients with elevated GAD (anxiety) showed a statistically significant elevated IBS severity. Thus, patients with anxious symptomatology seem not to be benefitting at all from an 8-week course of AMT therapy, suggesting that either this type of therapy is not suitable for anxious IBS patients or 8 weeks of AMT are too little for an effective improvement. Furthermore, it is quite possible that intervention modules such as gut hypnosis, which are used effectively in other studies with anxiety disorders [2, 17, 21], may have been implemented too weakly, too infrequently or not optimally by the young psychologists conducting the study. It is also possible that due to the suffering and fear of deteriorating IBS, patients develop greater symptomatic anxiety during the course of the disease. We therefore suggest screening for mental comorbidity regularly, and especially for depression and anxiety symptoms in IBS patients, directing them towards an effective CBT therapy. This stands in line with other studies which suggest the timely recognition of these patients and offering them a CBT treatment to improve the prognosis of IBS [7, 16, 30]. One recent study by Lackner et al. also showed interesting effects, namely that patients with anxiety disorders react differently to CBT and educational intervention [21].

The present study was limited by a small sample size and by a slight over-representation of women. This is in line with the comparison population in Germany, where significantly more women suffer from IBS (pooled odds ratio at 1.46) and women in the second and third decade of life are usually diagnosed twice as often [5]. Furthermore, we did not perform a comparison to a control or waiting group. In further research with this therapy concept, a waiting group design should be implemented in the future. However, this limitation has less influence on the research question of the difference in the severity of irritable bowel syndrome in patients without and with mental comorbidity than on the pure effectiveness of AMT as such. All patients included in this study were recruited by inviting former patients of the hospital GKH, of

medical care centers and of medical practices in the Berlin-Brandenburg region that are regularly working together with the hospital GKH. It is possible that patients with higher interest in anthroposophical medicine are overrepresented in this study due to their interest. The involvement of different individuals as therapists for each module (painting, sculpturing, gut hypnosis, nutritional counselling and cognitive training) may have had a further negative effect on individual aspects or modules of AMT. Even though patients with less than half or more missed days were excluded, another possible limitation is that patients who dropped out of the study may have worsened even more during the course of therapy than those that remained. However, there were still a number of patients who unfortunately worsened during the course of therapy. Here, in particular, the linear mixed effect analysis showed that the discrepancy in success occurred between the two groups with mental comorbidity and the patients without. However, this study served as a pilot and feasibility study and in a next step, these group comparisons between IBS patients with and without mental comorbidity and IBS patients with standard care or minimal care would and should be included in a larger study.

Our results suggest that the use of an 8-week AMT course is feasible and can have a profound and lasting effect in non-anxious and non-depressed irritable bowel syndrome patients. For our next study, we are planning a supportive, virtual three-month AMT refresher (either as an app, online tutorial, or a renewed face-to-face course) starting after the 8-week AMT program for IBS patients to practice at home. It has been reported that similar techniques, either with CBT, hypnotherapy, meditation or yoga, are learned skills that take time to show improvements in IBS-related symptoms [2, 7, 19, 30]. These refresher exercises, which are considered external motivators, are given to patients on prescription along with a patient diary in which patients are asked to document their IBS symptoms and health-related quality of life. Since IBS is often accompanied by anxiety and depression symptoms, an AMT concept tailored to these patients with additional psychotherapeutic modules such as individual and group therapy is highly advised but would need to be further evaluated in a future clinical trial. For future studies, it is strongly recommended to take into account the differentiation between both patient groups and to develop a more specific multimodal concept that is then tailored to both patient groups with and without mental comorbidity.

## Conclusions

Our findings reveal that an 8-week AMT concept is effective for IBS patients without mental disorders and that the reduction of severity symptoms has been maintained during the 12 month follow-up period. As IBS severity is highly correlated with mental comorbidity (depression and anxiety), those patients should be screened and be offered an effective and strong psychotherapeutic module prior to AMT, parallel to AMT or any other intended IBS therapy. In the future, it has to be determined whether such a tailored AMT concept may be effective for depressive or anxious IBS patients. In addition, the interaction between depression, anxiety and IBS severity over the course of time has to be further investigated.

## Supporting information

**S1 Checklist.**
(DOC)

**S1 File.**
(PDF)

**S2 File.**
(PDF)

## Acknowledgments

We would like to thank Melisa Celik (MC), Matthias Hunklinger (MAH), Anna Kruschel (AK), Liliane Ludin (LL) and Louisa Wagner (LW) for their great help in the implementation of the study. The students carried out their Master's theses in clinical psychology at Steinbeis University Berlin during the study. Besides the general help in the implementation, LL and LW were involved in the cognitive training group, AK and MAH in the gut hypnosis, LW and LW in the psychoeducation and MAH and AK with the external applications (wraps) of AMT and carried them out according to the study protocol. MC conducted the 12-month follow-up assessment.

## Author Contributions

**Conceptualization:** Harald Matthes.

**Formal analysis:** Maximilian Hinse.

**Funding acquisition:** Harald Matthes.

**Investigation:** Maximilian Hinse, Harald Matthes.

**Methodology:** Maximilian Hinse, Harald Matthes.

**Project administration:** Maximilian Hinse, Harald Matthes.

**Supervision:** Anja Thronicke, Anne Berghöfer, Harald Matthes.

**Visualization:** Maximilian Hinse.

**Writing – original draft:** Maximilian Hinse.

**Writing – review & editing:** Maximilian Hinse, Anja Thronicke, Anne Berghöfer, Harald Matthes.

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
