## [Decision Letter · Decision Letter 0]

8 Jun 2022

PONE-D-22-06021New multimodal intervention to reduce Irritable Bowel Syndrome (IBS) severity symptoms - pilot study with a 12 month’s follow upPLOS ONE

Dear Dr. Hinse,

Thank you for submitting your manuscript to PLOS ONE. After careful consideration, we feel that it has merit but does not fully meet PLOS ONE’s publication criteria as it currently stands. Therefore, we invite you to submit a revised version of the manuscript that addresses the points raised during the review process.

We look forward to receiving your revised manuscript.

Kind regards,

Wan-Long Chuang, M.D., Ph.D.

Academic Editor

PLOS ONE

Journal Requirements:

3. The in-house editorial staff feels that your study meets the World Health Organization definition of a clinical trial because it is a prospective study in which participants received 12 session of AMT to investigate the effects on IBS.

Reviewers' comments:

Reviewer's Responses to Questions

**Comments to the Author**

1. Is the manuscript technically sound, and do the data support the conclusions?

Reviewer #1: Yes

Reviewer #2: Partly

Reviewer #3: Yes

2. Has the statistical analysis been performed appropriately and rigorously? 

Reviewer #1: Yes

Reviewer #2: Yes

Reviewer #3: Yes

3. Have the authors made all data underlying the findings in their manuscript fully available?

Reviewer #1: Yes

Reviewer #2: Yes

Reviewer #3: Yes

4. Is the manuscript presented in an intelligible fashion and written in standard English?

Reviewer #1: Yes

Reviewer #2: No

Reviewer #3: Yes

5. Review Comments to the Author

Reviewer #1: Important note: This review pertains only to ‘statistical aspects’ of the study and so ‘clinical aspects’ [like medical importance, relevance of the study, ‘clinical significance and implication(s)’ of the whole study, etc.] are to be evaluated [should be assessed] separately/independently. Further please note that any ‘statistical review’ is generally done under the assumption that (such) study specific methodological [as well as execution] issues are perfectly taken care of by the investigator(s). This review is not an exception to that and so does not cover clinical aspects {however, seldom comments are made only if those issues are intimately / scientifically related & intermingle with ‘statistical aspects’ of the study}. Agreed that ‘statistical methods’ are used as just tools here, however, they are vital part of methodology [and so should be given due importance].

COMMENTS: Please note that, your ABSTRACT is well drafted but assay type. It is preferable [refer to item 1b of CONSORT checklist 2010: Structured summary of trial design, methods, results, and conclusions] to divide the ABSTRACT with small sections like ‘Objective(s)’, ‘Methods’, ‘Results’, ‘Conclusions’, etc. which is an accepted practice of most of the good/standard journals [including this one, though ‘The PLoS One Guidelines to Authors’ did not specify an Abstract format, it is desirable]. It will definitely (generally) be more informative then, I guess, whatever the article type may be.

This study being ‘pilot’ in nature, sample size is not a big issue. However, [though many things are ignored (loosely looked at / evaluated)] methodological issues need to be very rigorous followed {in case of clinical trial, CONSORT guidelines are to be strictly observed/followed}. I am sure that the authors are aware of the well-known drawbacks of a single-arm design [a type of Quasi-experimental research], and it is often said that ‘alright to have ‘single-arm design’ (before-after study) when that is the only possibility’, however, it is very essential to keep the limitations in mind while interpreting results. Considering ‘pilot’ (a feasibility study) nature of the study although it is acceptable/alright, as you know, a classical/ideal clinical trial/study needs/requires a concurrently {but similarly} handled/treated appropriately selected/chosen control/comparison parallel group/arm. In this context, please read the following which is pasted from one standard textbook on ‘Research Methodology’:

“Inferential statistics (i.e., hypothesis testing + estimation of CI) is built on the population model [which means the underlying assumption is that there is/are population(s) and we are dealing with random sample(s) drawn from that/those population(s)]. Although in clinical trial (involving at least two groups) we do not really deal with random samples (generally a non-probabilistic convenience sampling), ‘allocation’ to treatment groups is ‘randomly’ done which enable us to evoke the population model and we can use inferential statistics safely. But when there is only one group (so that there is no question of random allocation), with ‘non-random’ selection, it may be questionable to use inferential statistics even if you have two measurement sets as ‘pre-post’ or use ‘internal grouping for comparison.”

Therefore, I guess, it would have been nice to adhered to original plan [lines 89-90] of ‘a waiting group design’.

Although all statistical analyses are performed correctly, few table values [example: q-values (false discovery rate correction for multiple testing) in table-2]] are not explained adequately, in my opinion. How to read [and interpret] these table values may be very briefly explained. Same with table-3.

Please note that the measures/tools used are definitely appropriate, but most of them yield data that are in [at the most] ‘ordinal’ level of measurement [and not in ratio level of measurement for sure. Then application of suitable non-parametric test(s) is/are indicated/advisable [even if distribution may be ‘Gaussian’ (i.e. normal)]. Agreed that there is/are no non-parametric test(s)/technique(s) available to be used as alternative in all situation(s) [suitable / most desired/applicable], but should be used whenever/wherever they are available [for example, instead of paired ‘t’ test use Wilcoxon’s Signed Ranked test].

Moreover, limitations (if any) of the study are not mentioned/listed anywhere. Does that mean {according to authors} there are none? As pointed out in ‘important note’ above “This review pertains only to ‘statistical aspects’ of the study and so ‘clinical aspects’ should be assessed separately/independently [one should carefully consider/look at the clinical implications of the study].

There is no problem in accepting this article after minor revision, in my opinion.

Reviewer #2: 1. The study population should be more specifically characterized so that we could know what the target population of AMT would be. What are the predominant symptoms? How about the subgroups (IBS-C, IBS-D, mixed…)? Did these bowel symptoms improve after AMT?

2. In this study, female patients were predominant (92% in abstract and 89% in main text, please clarify) so the generalizability of this study should be discussed in more depth. Traditionally, male IBS patients tend to have diarrhea (IBS-D) and female patients are more likely to have constipation.

3. The novelty of this study is AMT. However, several of the modules and sessions of AMT were conducted by psychology students. Did these students receive related medical training regarding IBS? Did they conduct the AMT under surveillance? What were their experiences and background? It would be better to clarify these points before AMT is put into clinical practice.

4. Similarly, the nutritional counselling and education for food tolerances were conducted by “experienced therapist”. Who were they? Background of nutrition?

5. Did these patients receive medical treatment during the AMT period? One patient received treatment for psychosis on enrolment? Should this patient be excluded?

6. Since only 21 of the 36 patients had the 12-month follow-up, it may not be very confident to say the AMT effect to be profound and lasting.

6. English editing by a native speaker is suggested.

Reviewer #3: Irritable bowel syndrome treatment is various and broad. It means that efficacy of single irritable bowel syndrome treatment is inadequate. Multimodal therapy to improve the severity of IBS in patients without psychological comorbidities with a sustained effect at 12 months’ follow-up in this study.

However there are some biases. First, apart from small sample size, the gender composition is not equilibrium and up to 92% female. Second, AMT was evaluated as a therapy package including physician-patient consultations, the question of specific therapy effects versus non-specific effects (placebo effects, context effects, patient expectations etc.) was not an issue of the present analysis. Third, 51 patients were included in the study but the dropout rate was 29.4%(15/51).

6. PLOS authors have the option to publish the peer review history of their article (what does this mean?). If published, this will include your full peer review and any attached files.

Reviewer #1: No

Reviewer #2: No

Reviewer #3: No

---

## [Author Response · Author response to Decision Letter 0]

30 Jun 2022

Thank you for giving us the opportunity to revise our manuscript and thank you very much for your valuable feedback and taking time in reviewing the manuscript.

Within our submitted revised manuscript we have tried to adhere to all remarks of the editors and reviewers, and have accordingly substantially revised the manuscript. Our detailed response to every point raised is in the attached file "Response to Reviewers".

---

## [Decision Letter · Decision Letter 1]

22 Aug 2022

PONE-D-22-06021R1New multimodal intervention to reduce Irritable Bowel Syndrome (IBS) severity symptoms - pilot study with a 12 month’s follow upPLOS ONE

Dear Dr. Hinse,

Thank you for submitting your manuscript to PLOS ONE. After careful consideration, we feel that it has merit but does not fully meet PLOS ONE’s publication criteria as it currently stands. Therefore, we invite you to submit a revised version of the manuscript that addresses the points raised during the review process.

We look forward to receiving your revised manuscript.

Kind regards,

Wan-Long Chuang, M.D., Ph.D.

Academic Editor

PLOS ONE

Journal Requirements:

Reviewers' comments:

Reviewer's Responses to Questions

**Comments to the Author**

1. If the authors have adequately addressed your comments raised in a previous round of review and you feel that this manuscript is now acceptable for publication, you may indicate that here to bypass the “Comments to the Author” section, enter your conflict of interest statement in the “Confidential to Editor” section, and submit your "Accept" recommendation.

Reviewer #1: All comments have been addressed

Reviewer #2: All comments have been addressed

Reviewer #4: All comments have been addressed

2. Is the manuscript technically sound, and do the data support the conclusions?

Reviewer #1: (No Response)

Reviewer #2: (No Response)

Reviewer #4: Partly

3. Has the statistical analysis been performed appropriately and rigorously? 

Reviewer #1: (No Response)

Reviewer #2: (No Response)

Reviewer #4: Yes

4. Have the authors made all data underlying the findings in their manuscript fully available?

Reviewer #1: (No Response)

Reviewer #2: (No Response)

Reviewer #4: Yes

5. Is the manuscript presented in an intelligible fashion and written in standard English?

Reviewer #1: (No Response)

Reviewer #2: (No Response)

Reviewer #4: No

6. Review Comments to the Author

Reviewer #1: COMMENTS: Since all of the comments made on earlier draft are considered positively, I recommend the acceptance because the manuscript now has achieved acceptable level, in my opinion.

Reviewer #2: (No Response)

Reviewer #4: The manuscript demonstrated an 8-week AMT therapy would improve the IBS and maintain its effect at after 12 months’ follow-up, which is limited in the IBS patients without psychological comorbidities. The manuscript has been received one round review by 3 reviewers. And I found the manuscript is interesting and the authors have adequately responded to the 3 reviewers. Nevertheless, I still have some other comments which need to be clarified before final publication.

1. The manuscript should be edited by a native English speaker before publication.

2. For most readers, even among IBS specialists, are unfamiliar with anthroposophic therapy. More detailed introduction regarding the background, rationale and efficacy in treating other disorders by using such therpay should be described in the INTRODUCTION.

3. As described in the Method, the AMT included psychoeducation, psychotherapy, art therapy, external application, and nutritional counselling. What is the rationale for including external application for IBS? What kind of nutrition suggestion was applied in the AMT? A more detailed description is required.

4. Since psychotherapy and nutritional suggestions have already been shown to be effective in reduction IBS severity, the current results suggested that the AMT by adding anthroposophic components may reverse the proven efficacy of psychotherapy/nutrition suggestions. The authors should comment the limitation on this.

5. Since multimodalities were included in AMT, did all the modalities were required to use by the IBS patients during the 8-week treatment period and follow-up? Was it possible that some IBS patients would prefer some modalities without using other modules, which would lead to the unfavorable outcome in the IBS patients with psychological co-morbidity?

7. PLOS authors have the option to publish the peer review history of their article (what does this mean?). If published, this will include your full peer review and any attached files.

Reviewer #1: **Yes: **Dr. Sanjeev Sarmukaddam

Reviewer #2: No

Reviewer #4: No

---

## [Author Response · Author response to Decision Letter 1]

5 Oct 2022

Reviewer #4: The manuscript demonstrated an 8-week AMT therapy would improve the IBS and maintain its effect at after 12 months’ follow-up, which is limited in the IBS patients without psychological comorbidities. The manuscript has been received one round review by 3 reviewers. And I found the manuscript is interesting and the authors have adequately responded to the 3 reviewers. Nevertheless, I still have some other comments which need to be clarified before final publication.

1. The manuscript should be edited by a native English speaker before publication.

ANSWER: Thank you for this suggestion. The manuscript has now been edited by a native speaker..

2. For most readers, even among IBS specialists, are unfamiliar with anthroposophic therapy. More detailed introduction regarding the background, rationale and efficacy in treating other disorders by using such therpay should be described in the INTRODUCTION.

ANSWER: Thank you for this suggestion. In order to describe the anthroposophic medicine and the concept behind the therapy of irritable bowel syndrome more comprehensibly for the readers, a subheading and an extended passage on anthroposophic medicine has been inserted from line 77 onwards.

3. As described in the Method, the AMT included psychoeducation, psychotherapy, art therapy, external application, and nutritional counselling. What is the rationale for including external application for IBS? What kind of nutrition suggestion was applied in the AMT? A more detailed description is required.

ANSWER: Thank you for this suggestion. However, our AMT concept included psychoeduction but notpsychotherapy. External applications were used as a self-help technique and served the patients as a method for relaxation and as a support for their self-healing powers. The patients got to know different oils in order to find a suitable self-help for themselves and later applied it independently at home. The aim of Nutritional counselling was to give patients education about food intolerances and different food groups and their impact on their microbiota and IBS. The experienced nutritionist did not give advice on specific diets but gave education and answered individual questions in four sessions. The descriptions of all modules have been revised from page 6, line 127 onwards to make them easier to understand.

4. Since psychotherapy and nutritional suggestions have already been shown to be effective in reduction IBS severity, the current results suggested that the AMT by adding anthroposophic components may reverse the proven efficacy of psychotherapy/nutrition suggestions. The authors should comment the limitation on this.

ANSWER: AS our AMT did not include a classical psychotherapy, and also because the subgroup patients with mental comorbidity did not improve after the AMT intervention, our suggestion and conclusion in the manuscript was to apply a psychotherapy for these patients either before AMT or during AMT session. Our study shows that especially for patients without mental comorbidity, the AMT therapy can have a great effect even without explicit psychotherapy. Therefore, it generally seems to make a lot of sense in IBS patients to screen them for their comorbidity and offer adapted therapy, e.g. including psychotherapy in addition to AMT. Lackner et al. (2019) also show that IBS patients with e.g. concurrent anxiety symptoms respond differently to CBT or only educational therapy.

5. Since multimodalities were included in AMT, did all the modalities were required to use by the IBS patients during the 8-week treatment period and follow-up? Was it possible that some IBS patients would prefer some modalities without using other modules, which would lead to the unfavorable outcome in the IBS patients with psychological co-morbidity?

ANSWER: All patients who participated in the study also participated equally in all modalities. Not all patients liked all modules equally (see page 18, line 321). In the satisfaction surveys and oral reports during the therapy, differences in satisfaction with the individual modules were observed. In the explorative statistical analysis, however, no correlations were found between the patients' satisfaction with the AMT therapy and the primary outcome, symptom improvement. These statistical results were not reported due to the focus of the publication and the lack of correlations.

---

## [Decision Letter · Decision Letter 2]

6 Nov 2022

New multimodal intervention to reduce Irritable Bowel Syndrome (IBS) severity symptoms - pilot study with a 12 month follow up

PONE-D-22-06021R2

Dear Dr. Hinse,

We’re pleased to inform you that your manuscript has been judged scientifically suitable for publication and will be formally accepted for publication once it meets all outstanding technical requirements.

Kind regards,

Wan-Long Chuang, M.D., Ph.D.

Academic Editor

PLOS ONE

Additional Editor Comments (optional):

Reviewers' comments:

Reviewer's Responses to Questions

**Comments to the Author**

1. If the authors have adequately addressed your comments raised in a previous round of review and you feel that this manuscript is now acceptable for publication, you may indicate that here to bypass the “Comments to the Author” section, enter your conflict of interest statement in the “Confidential to Editor” section, and submit your "Accept" recommendation.

Reviewer #1: All comments have been addressed

Reviewer #2: All comments have been addressed

Reviewer #4: All comments have been addressed

2. Is the manuscript technically sound, and do the data support the conclusions?

Reviewer #1: (No Response)

Reviewer #2: Yes

Reviewer #4: Yes

3. Has the statistical analysis been performed appropriately and rigorously? 

Reviewer #1: (No Response)

Reviewer #2: Yes

Reviewer #4: Yes

4. Have the authors made all data underlying the findings in their manuscript fully available?

Reviewer #1: (No Response)

Reviewer #2: Yes

Reviewer #4: Yes

5. Is the manuscript presented in an intelligible fashion and written in standard English?

Reviewer #1: (No Response)

Reviewer #2: Yes

Reviewer #4: Yes

6. Review Comments to the Author

Reviewer #1: COMMENTS: Since all of the comments made on earlier draft are considered positively, I recommend the acceptance because the manuscript now has achieved acceptable level, in my opinion. In fact, the earlier version was already accepted on 22.07.2022.

Reviewer #2: The authors have adequately responded to the reviewers' comments and suggestions. I think this manuscript is ready for publication.

Reviewer #4: The authors have addressed all my concerns I raised. And I am quite satisfied with the authors' answers.

7. PLOS authors have the option to publish the peer review history of their article (what does this mean?). If published, this will include your full peer review and any attached files.

Reviewer #1: **Yes: **Dr. Sanjeev Sarmukaddam

Reviewer #2: No

Reviewer #4: No

---

## [Editor Report · Acceptance letter]

9 Nov 2022

PONE-D-22-06021R2 

New multimodal intervention to reduce Irritable Bowel Syndrome (IBS) severity symptoms - pilot study with a 12 month follow-up 

Dear Dr. Hinse:

I'm pleased to inform you that your manuscript has been deemed suitable for publication in PLOS ONE. Congratulations! Your manuscript is now with our production department. 

Kind regards, 

on behalf of

Dr. Wan-Long Chuang 

Academic Editor

PLOS ONE